# Who Wants to (Digitally) Live Forever? The Connections That Narcissism Has with Motives for Digital Immortality and the Desire for Digital Avatars

**DOI:** 10.3390/ijerph20176632

**Published:** 2023-08-23

**Authors:** Avi Besser, Tal Morse, Virgil Zeigler-Hill

**Affiliations:** 1Department of Communication Disorders, Hadassah Academic College, Jerusalem 91010, Israel; 2Department of Photographic Communication and The Centre for Death and Society, University of Bath, Bath BA2 7AY, UK; talmor@hac.ac.il; 3Department of Psychology, Oakland University, Rochester, MI 48309, USA

**Keywords:** narcissism, digital immortality, digital avatars, death anxiety

## Abstract

We examined the role that death anxiety (for self and others) and motivation for digital immortality played in the associations that narcissistic personality traits had with the desire for digital avatars (of self and others) in a sample of Israeli community members (N = 1041). We distinguished between four forms of narcissism: extraverted narcissism (characterized by assertive self-enhancement), antagonistic narcissism (characterized by defensiveness and hostility), neurotic narcissism (characterized by emotional distress), and communal narcissism (characterized by attempts to emphasize superiority over others by exaggerating communal characteristics such as being extraordinarily helpful). Our sequential parallel mediation analyses showed that narcissistic personality traits were associated with fear of death and the desire for symbolic immortality (having a digital avatar for self and others), with mainly indirect associations via fear of death and the motivation for eternal life and to be there for others. Discussion is focused on the role that fear of death and specific “defensive control” motives for having digital avatars (e.g., motivation for eternal life and to be there for others) may play in the desire for digital immortality reported by individuals with narcissistic personality traits.

## 1. Introduction

Digital avatars are rapidly increasing in prevalence and complexity. These digital avatars provide representations of their owners in various contexts, including immersive video games and virtual reality simulations. In recent years, there has been a growing interest in the possibility that digital avatars may continue to provide representations of individuals following their deaths. This is an important issue because death is, of course, an inevitable part of life. As a result, issues surrounding death—such as how we balance our desire to live with the certainty that we will eventually die—constitute an existential dilemma for all of us [1]. The possibility of maintaining some form of existence beyond death has tremendous appeal, which may explain why various religions contain some form of an “afterlife” [2]. Technology may eventually give rise to something akin to a new form of life after death: “digital immortality.” That is, individuals may eventually be able to create and maintain a digital presence by uploading a copy of their personality to a digital medium that will allow them to “outlive” their physical selves [3,4,5]. This might sound like science fiction, but our current level of technology may soon allow for the creation and activation of highly sophisticated digital avatars that will draw on photography and deep-fake technologies to look like a real person, utilize voice sampling to sound like a real person, and take advantage of recent advances in artificial intelligence (AI) to, arguably, think like a real person. In essence, these digital avatars would provide users with something akin to a form of digital immortality.

The primary focus of the present study was to provide some initial information regarding which individuals would have the greatest desire for this sort of digital immortality through the use of digital avatars. More specifically, we were interested in the connections that narcissistic personality traits would have with the desire for digital avatars and the possibility that the fear of death and motives for digital avatars may play roles in these associations. The findings suggest that motives for the desire for a digital avatar for self and others are similar to a great extent; however, there are additional indirect associations that can explain the desire for avatars for others, which means that there are alternative mediators for that desire.

### 1.1. The Development of Digital Avatars

Narcissistic personality traits have received considerable attention in the literature due to their associations with various problematic outcomes, including aggression, grandiosity, and an inflated sense of self-importance [6,7]. With the proliferation of social media and other digital platforms, individuals with narcissistic personality traits have been shown to use these forms of technology to engage in self-promotion and seek out opportunities for self-enhancement [8,9]. Despite the considerable literature concerning narcissism and recent interest in issues surrounding the use of digital avatars, little is known about how these two phenomena may intersect. For example, it is unclear whether narcissistic personality traits would increase the appeal of digital avatars by providing something akin to a form of digital immortality or whether these digital possibilities are mitigated or enhanced by the fear of death.

The use of cyberspace to accommodate digital practices of interaction with death dates back to the 1990s [10]. In its early forms, people used the Internet to establish digital cemeteries and memorial websites [11]. The internet allowed new forms of engagement with death and proposed the promise of digital immortality. When social networks became prevalent, bereaved people found comfort in utilizing online platforms for mourning and commemoration [12]. The literature on online mourning and digital commemoration has thus far pointed to two types of practices that employ digital platforms for these matters: (1) grief-specific sites that were generated by the bereaved after the deceased’s death to cultivate mourning spaces; and (2) non-grief-specific sites that posthumously accommodate the online content the now-dead-users posted when they were alive. Whether these are grief-specific sites or non-grief-specific sites that were converted into memorial sites, these sites are usually initiated by the bereaved family and friends for mourning and commemoration. As such, they do not reflect a desire to perpetuate the self but rather to commemorate the other.

Furthermore, these sites are unidirectional communication platforms that enable the bereaved to communicate with the dead or with other members of the bereaved community, but they do not facilitate two-way communication between the living and the digital representation of the dead [3]. Thus, since these commemoration platforms entertain immortality, the deceased and memorialized users are dormant and are unable to respond and interact with their loved ones. In recent years, new possibilities for digital mourning and commemoration have emerged. Namely, the creation of digital avatars that would respond to and interact with the bereaved is coming to fruition. These ventures provide individuals with a semblance of digital immortality.

Digital avatars that resemble human behavior take advantage of recent technological developments in the fields of human-computer interactions (HCI) and artificial intelligence (AI). Voice-activated virtual assistants—such as Siri and Alexa—are commonplace, and people have become accustomed to speaking to digital machines. Moreover, Google Duplex, an AI virtual assistant, was able to make phone calls to reserve a table at a restaurant without the restaurant employee realizing they were communicating with a computer [13]. In November 2022, OpenAI launched ChatGPT (i.e., an AI chatbot) that not only retrieves information in a friendly and accessible manner but also responds like a human in text-based conversations. This application became the fastest-growing consumer software application in history [14].

Virtual assistants are designed to behave like humans, but they do not mimic a specific person. In contrast, the digital afterlife industry (DAI) aims to harness these technological developments to create digital “clones” of specific people. In February 2021, Microsoft patented a chatbot that would allow users to interact with digital avatars of dead users. Shortly after the announcement, Microsoft said that they currently have no plans to release this product [15], but other companies like Eter9 [16], Lifenaut (2017) [17], and Eternime (2017) [18] are already focused on creating digital avatars that mimic the personalities of specific individuals so that they can create and maintain a digital presence that can persist after their death (see [3] or [19], for reviews).

Like the creation of commemoration sites, the motivation to pursue such digital immortality is often initiated by the bereaved rather than by the deceased, regardless of their explicit consent. MyHeritage has launched its Deep Nostalgia application, which creates short-moving GIFs based on photographs of dead ancestors (MyHeritage). In Israel, two different organizations utilized deep-fake technologies to “resurrect” dead people in national TV campaigns [20], and two dead singers were brought together for an AI recording of a song that was written this year [21]. Amazon is developing a feature that will use voice samplings to replace Alexa’s voice with that of living or dying people [22]. In South Korea, a TV production has created a 3D avatar of a deceased young girl to allow her mother to interact with her one more time using a VR headset and haptic gloves [23]. Actor Joshua Borbeau created an AI version of his fiancée by uploading her digital remains into an early version of ChatGPT ([24]; see also [25]). More recently, Paul McCartney announced that he will soon release a new Beatles song, using AI to recreate the voices of the deceased band members [26].

While the above examples were not initiated by the deceased, other projects draw on the consent and collaboration of people that will embody the avatar, driven by the motivation to facilitate collective memory and personal commemoration. The Shoah Foundation recruited Holocaust survivors who were filmed testifying about their personal stories to allow conversations with their holograms after they die (Dimensions in Testimony). Journalist James Vlahos filmed his dying father, curated content about him, and later created the *Dad-bot*, a chatbot of his dead father with whom he continues to communicate [27]. These examples serve the construction of collective and family memory, and they manifest a self-desire to become a digital avatar, but the personal motivation to partake in these projects—whether altruistic, narcissistic, or other—was never studied.

The possibility that technology may eventually be able to provide a form of digital immortality has already become a growing concern in contemporary society because of the various questions that it raises. While enterprises like Dimensions in Testimony and Deep Nostalgia can play a significant positive role in cultivating collective memory or commemorating ancestors, the full effect of digital immortality on mourning and bereavement is yet to be studied [28]. Grief and mourning scholars have explored the use of digital technologies to facilitate continuing bonds and soothe the difficulties of coping with loss, but some find comfort in digital interactions with the dead (e.g., [29,30]). However, engagement with digital representations and manifestations of the dead is often perceived as unpleasant and disturbing by others [31].

Besides clinical and therapeutic concerns, there are legal and ethical issues pertaining to post-mortem privacy and control over orphaned online profiles [32,33] and issues surrounding how to potentially create digital copies of a human personality, who is entitled to a post-mortem avatar (and who is not), what happens if an avatar becomes sentient or disloyal to the deceased’s personality, and who has the right to terminate an avatar. These issues—along with many others—have also been receiving attention from scholars in various fields (e.g., [19,34,35,36,37]).

Although various motives have been discussed for the desire for digital immortality (e.g., living on, continuing bonds; [3,19]), no specific individual differences have been examined in conjunction with the desire to have digital avatars. A recent study has found that some demographic characteristics and online activities can explain a tendency to leave a post-mortem avatar [31], but that study did not explore personality traits. The present study is the first to examine which personality traits can explain both the desire of individuals to have a digital avatar based on themselves that could continue to communicate with their family members and friends after their death and the desire to have digital avatars based on some of their close family members and friends so they could continue to communicate with these digital representations posthumously. We believe that personality traits may help us understand which individuals would be most interested in the sort of digital immortality offered by the use of digital avatars. More specifically, we think that narcissism may be connected with the desire for these digital avatars.

### 1.2. Narcissism and the Desire for Digital Immortality

It is unclear whether narcissistic individuals would have a particularly strong desire for the sort of digital immortality offered by the use of digital avatars. One argument in favor of this sort of connection is offered by self-expansion theory, which argues that individuals tend to be motivated to expand their sense of self through strategies such as gaining resources, increasing their competencies, and broadening their perspectives [38]. In fact, narcissism is positively associated with a spatial-symbolic form of self-expansion (i.e., the use of external objects as identity markers that could be used to expand the self) [39]. This aligns with the argument that individuals can impart meaning to external objects that serve as symbols that allow for the expansion of self beyond one’s physical body [40,41]. Digital avatars may serve this sort of role and allow individuals to expand beyond their own physical bodies, and these symbols of the expanded self may even continue to function after the person’s physical body has died. If there were connections between narcissistic personality traits and the desire to use digital avatars to attain a form of digital immortality, then factors such as the fear of death and motives for digital immortality could play roles in these connections. More specifically, we believed that the fear of death and motives for employing digital avatars may serve as mechanisms to help us understand any connections that exist between narcissistic personality traits and the desire for digital avatars.

Fear of death—or death anxiety—is defined as the anxiety related to awareness of one’s own mortality and inevitable death [42]. Fear of death is associated with a wide array of outcomes, including safety-related concerns and behaviors [43] and religiosity [44]. There has been a great deal of speculation regarding the connection that narcissism has with the fear of death [45]. One common idea is that certain narcissistic characteristics, such as the pursuit of self-importance, may be an attempt to protect individuals from the fear of death [46,47]. This suggests that individuals with narcissistic personality traits may be so deeply afraid of dying that they pursue various strategies to prolong their lives and evade death (e.g., gaining fame, acquiring wealth, denying the inevitability of death). Although there have been relatively few empirical studies concerning the connection between narcissism and the fear of death, the general conclusion from the existing studies is that narcissistic individuals report a greater fear of death than other individuals [48,49]. There is considerable evidence that thoughts of one’s own death (i.e., mortality salience) may promote the use of various strategies to strive for literal or symbolic forms of immortality [50,51,52,53]. Further, there is some evidence that personality traits linked with narcissism (e.g., entitlement) may enhance reactions to situations involving mortality salience [54,55], but it is important to note that similar effects have not emerged in other studies [49].

Narcissism is often conceptualized as an agentic construct [56], but it has been argued that there may also be a form of communal narcissism [57]. Communal narcissism is focused on the satisfaction of motives that are similar to those of grandiose narcissism (e.g., status, grandiosity, esteem, power), but individuals with elevated levels of communal narcissism attempt to satisfy these motives through the exaggeration of communal qualities (e.g., portraying themselves as being exceptionally helpful and kind) rather than agentic qualities (e.g., portraying themselves as being intelligent; [57]). Given its other similarities to agentic narcissism [58], communal narcissism may have similar connections with the fear of one’s own death (e.g., total isolation of death, shortness of life, never thinking or experiencing again) along with concerns regarding the death of others (e.g., losing someone close, never being able to communicate again, feeling lonely without the person). It is also possible that communal narcissism would be associated with the motivation for a form of digital immortality through the use of digital avatars for self and close others.

### 1.3. Overview and Predictions

Our aim for this study was to consider the connections that narcissistic personality traits had with the desire to acquire a form of digital immortality through the use of digital avatars of themselves and others. In addition, we considered the possibility that these associations would be mediated by fear of death and various motives for having digital avatars. By exploring these issues, we hoped to shed additional light on how individuals with narcissistic personality traits may approach the concept of digital immortality and how this may be influenced by their underlying fears concerning death and their motives for seeking digital immortality. Previous research concerning narcissism and issues surrounding death has often employed a unidimensional view of narcissism [48,49], but there is considerable evidence that narcissism is actually a multidimensional personality construct [59,60]. One popular model argues that there are three distinct narcissistic personality traits [61]: extraverted narcissism (which involves assertive self-enhancement), antagonistic narcissism (which involves hostility and self-protection), and neurotic narcissism (which involves emotional distress). In addition, we also included communal narcissism (which involves attempts to emphasize superiority over others by exaggerating communal characteristics such as being extraordinarily helpful; [57]) in order to gain an even broader understanding of the connections that narcissistic personality traits have with issues concerning the desire for digital immortality.

We believed it was important for us to distinguish between these four aspects of narcissism when considering the connections that narcissistic personality traits had with the desire for digital immortality through the use of digital avatars. This approach is consistent with the recognition that it is important for researchers to distinguish between different aspects of narcissism in order to develop a more complete and nuanced understanding of the attitudes and behaviors that characterize narcissism. For example, these different aspects of narcissism tend to have divergent associations with various motives and behaviors [62,63]. A model depicting the proposed associations that narcissistic personality traits may have with the desire to have digital avatars of self and others through fear of death and motives for having digital avatars is presented in Figure 1. As shown in the model, we believed that narcissistic personality traits would be associated with the fear of death for self and others. In turn, we expected the fear of death to be associated with motives for employing digital avatars (i.e., eternal life, legacy, and being there for others). Finally, we anticipated that these motives for employing digital avatars would be positively associated with the desire for digital avatars of self or others.

We developed the following hypotheses for this study:

**Hypothesis 1a:** *Extraverted narcissism will be positively associated with the desire for digital avatars of self and others*.

**Hypothesis 1b:** *We expected each of the motives for having digital avatars to mediate the associations that extraverted narcissism had with the desire for digital avatars of self and others*.

**Hypothesis 1c:** *We did not have clear hypotheses about whether the fear of death for self or others would further mediate the associations that extraverted narcissism had with the desire for digital avatars of self and others through the motives for having digital avatars. However, since death-related concerns have been shown to promote the use of strategies involved with either literal or symbolic forms of immortality [50,51,52,53], we examined the potential role that the fear of death had in these associations for exploratory purposes*.

**Hypothesis 2a:** 
*Antagonistic narcissism will be positively associated with the desire for a digital avatar for self, but it will be negatively associated with the desire for digital avatars of others.*


**Hypothesis 2b:** 
*We expected the motivation for eternal life and the motivation for legacy to mediate the associations that antagonistic narcissism had with the desire for a digital avatar for self. We did not have clear hypotheses about whether each of the motives for having digital avatars would mediate the association that antagonistic narcissism had with the desire for digital avatars of others, but we examined those indirect associations for exploratory purposes.*


**Hypothesis 2c:** *We expected the fear of death for self to further mediate the associations that antagonistic narcissism had with the desire for a digital avatar for self through the motivation for eternal life and the motivation for legacy. That is, we anticipated that antagonistic narcissism would be positively associated with the fear of death for self, which would be positively associated with the motivation for eternal life and the motivation for legacy. In turn, these motives would be positively associated with the desire to have a digital avatar for self. We did not have clear hypotheses about whether the fear of death would mediate the associations that antagonistic narcissism had with the desire for digital avatars of others, but we examined those indirect associations for exploratory purposes*.

**Hypothesis 3a:** *Neurotic narcissism will be positively associated with the desire for digital avatars of self and others*.

**Hypothesis 3b:** *We expected each of the motives for having digital avatars to mediate the associations that neurotic narcissism had with the desire for digital avatars of self and others*.

**Hypothesis 3c:** *We expected the fear of death for self and others to further mediate the associations that neurotic narcissism had with the desire for digital avatars for self and others through the motives for having digital avatars. That is, we anticipated that neurotic narcissism would be positively associated with the fear of death, which would be positively associated with the motives for having digital avatars. In turn, the motives for having digital avatars would be positively associated with the desire to have digital avatars of self and others*.

**Hypothesis 4a:** *Communal narcissism will be positively associated with the desire for digital avatars of self and others*.

**Hypothesis 4b:** *We expected each of the motives for having digital avatars to mediate the associations that communal narcissism had with the desire for digital avatars of self and others*.

**Hypothesis 4c:** *We did not have clear hypotheses about whether the fear of death for self or others would further mediate the associations that communal narcissism had with the desire for digital avatars of self and others through the motives for having digital avatars. However, we examined the potential role that the fear of death had in these associations for exploratory purposes*.

## 2. Materials and Methods

### 2.1. Participants and Procedure

Participants in this study were 1081 Israeli community members who responded to requests asking for volunteers to take part in an online study via postings on various social media platforms and flyers that were placed in public locations. We excluded data from 40 participants due to them being univariate outliers (*n* = 32), providing inconsistent responses (*n* = 1), or having invariant response patterns (*n* = 7). The final 1041 participants (405 men and 636 women) had an average age of 32.24 years (*SD* = 13.39 [range = 18–87 years]). The mean number of years of education was 13.82 years (*SD* = 2.92), and the individuals who participated in the study were predominantly Jewish (91%). The self-reported current economic status of the participants was 20% “much higher than the minimum wage,” 33% “higher than the minimum wage,” 17% “similar to the minimum wage,” 17% “lower than the minimum wage,” and 13% “much lower than the minimum wage.” Slightly more than half of the participants reported losing a family member or friend during recent years (51%) and being actively engaged in online activities such as publishing content or frequently posting on social media sites (53%).

### 2.2. Measures

*Narcissism.* We used the short form of the Five-Factor Narcissism Inventory [64] to measure the following aspects of narcissism: *extraverted narcissism* (16 items; “I like being noticed by others” [α = 0.76]), *antagonistic narcissism* (32 items; “I hate being criticized so much that I can’t control my temper when it happens” [α = 0.85]), and *neurotic narcissism* (12 items; “When people criticize me, I get embarrassed” [α = 0.81]). Responses were provided using scales that ranged from 1 (*strongly disagree*) to 5 (*strongly agree*).

*Communal Narcissism.* The Communal Narcissism Inventory [57] was used to capture communal narcissism (16 items; e.g., “I am the most helpful person I know” [α = 0.90]). Responses were provided using scales that ranged from 1 (*disagree strongly*) to 5 (*agree strongly*).

*Fear of Death.* We used the Revised Collett–Lester Fear of Death Scale [65] to measure *fear of death for self* (8 items; “The total isolation of death” [α = 0.83]) and *fear of death for others* (8 items; “The loss of someone close to you” [α = 0.78]). Responses were provided using scales that ranged from 1 (*not at all*) to 5 (*very*).

*Motivation for Digital Avatar*. Participants were asked to read the following vignette concerning the development of digital avatars: *“Companies such as Microsoft and Eter9 are using artificial intelligence (AI) to develop “chatbots” for various situations. A chatbot is a service in which users can have conversations with a computer program as if they were communicating with a real person. One specific use of this AI technology is to develop chatbots that can simulate the responses of a specific person based on a combination of their responses to specific interview questions (e.g., Who is the most important person in your life? What do you think is your most valuable characteristic?) and stored data (e.g., email correspondence, WhatsApp conversations, Facebook posts, Instagram photos, Netflix watch lists, Spotify playlists). This technology would allow users to interact with chatbots that are designed to simulate the responses of a range of specific people, including historical figures, celebrities, and politicians. One intriguing direction for this technology is to develop chatbots that would be able to simulate the responses of individuals after they die. This would allow the family and friends of deceased individuals to interact with a chatbot that is based on their deceased loved one. For example, a young boy would be able to interact with a chatbot based on his recently deceased grandfather. The same chatbot could also be used by the young boy’s grandmother to help manage her loneliness after the death of her husband. In essence, this technology would allow these chatbots to serve as digital avatars for those who have died so that their family members and friends can find comfort in continuing to communicate with their deceased loved ones.”*

After reading the vignette, participants were asked about the extent to which they agreed with each of three motives for using a digital avatar: (1) achieve a form of “eternal life” even if it was only as a digital avatar; (2) leave a “legacy” so that others would have a way to remember them (e.g., stories); and (3) “be there” for others who may want to maintain some form of contact. Responses were provided using scales that ranged from 1 (*not at all*) to 7 (*very much*).

*Desire for Digital Avatars***.** Participants were asked about the extent to which they desired a digital avatar for themselves (i.e., “I would like to have a digital avatar based on me that could continue to communicate with my family members and friends after my death”) or for close others (i.e., “I would like to have digital avatars based on some of my family members and friends so that I could continue to communicate with them after their deaths”). Responses were provided using scales that ranged from 1 (*not at all*) to 7 (*very much*).

### 2.3. Data Analysis

We began analyzing the data using correlation coefficients to understand the zero-order correlations between narcissistic personality traits, fear of death, motives for digital avatars, and the desire for digital avatars. We followed these zero-order correlations with a series of sequential parallel multiple mediation analyses using a customized version of model 81 of the PROCESS macro [66]. We used these analyses because we expected narcissistic personality traits to be associated with the fear of death. Then, we expected the fear of death to be associated with motives for digital avatars, which, in turn, would be associated with the desire for digital avatars. These analyses allowed us to examine whether narcissistic personality traits had indirect associations with the desire for digital avatars through motives for digital avatars via fear of death. More specifically, we included narcissism as our predictor, fear of death (i.e., fear of death for self and fear of death for others) as our primary mediators, motives for digital avatars (i.e., eternal life, legacy, and being there for others) as our secondary mediators, and desire for digital avatars (i.e., desire for digital avatar of self and desire of digital avatars of others) as our outcomes. Because of their overlap, we were concerned that including the narcissistic personality traits in the same analysis might make it difficult to understand how they were related to the desire for avatars. As a result, we conducted separate analyses in which each narcissistic personality trait served as the predictor in its own model.

## 3. Results

Descriptive statistics and zero-order correlations are presented in Table 1. Extraverted narcissism was positively correlated with antagonistic narcissism, communal narcissism, fear of death for self and others, the motives for having digital avatars, and the desire to have digital avatars of self and others, whereas it was negatively correlated with neurotic narcissism. Antagonistic narcissism was positively correlated with communal narcissism, fear of death for self and others, the motives for having digital avatars, and the desire to have digital avatars of self and others, whereas it was not correlated with neurotic narcissism. Neurotic narcissism was positively correlated with fear of death for self and others, whereas it was negatively correlated with communal narcissism. Neurotic narcissism was not correlated with the motives for having digital avatars or the desire to have digital avatars of self or others. Communal narcissism was positively correlated with the fear of death for self and others, motives for having digital avatars, and the desire to have digital avatars of self and others. Fear of death for self and fear of death for others were positively correlated with the motives for having digital avatars and the desire to have digital avatars of self and others. The motives for having digital avatars were positively correlated with the desire to have digital avatars of self and others.

### 3.1. Desire for a Digital Avatar of Self

The results of the sequential parallel multiple mediation analyses for the desire for a digital avatar of self are presented in Table 2. These results showed that extraverted narcissism, antagonistic narcissism, and communal narcissism had positive total associations with the desire for a digital avatar of self that were mediated by various motives. For extraverted narcissism, it was the motivation to be there for others that mediated its association with the desire for a digital avatar of self. Antagonistic narcissism and communal narcissism had indirect associations with the desire for a digital avatar of self through fear of death for self via the motivation for eternal life. Communal narcissism also had another indirect association with the desire for a digital avatar of self through the fear of death for self via the motivation to be there for others. In contrast to the other narcissistic personality traits, neurotic narcissism did not have a significant total association with the desire for a digital avatar of self. However, it did have contrasting indirect associations with this desire. More specifically, neurotic narcissism had a negative indirect association with the desire for a digital avatar of self through the motivation for eternal life, whereas it had a positive indirect association with this desire through the fear of death for self via the motivation for eternal life. In addition, neurotic narcissism had a positive indirect association with the desire for a digital avatar of self through the fear of death for self via the motivation to be there for others.

### 3.2. Desire for a Digital Avatar of Others

The results of the sequential parallel multiple mediation analyses for the desire for digital avatars of others are presented in Table 3. These results showed that extraverted narcissism, antagonistic narcissism, and communal narcissism had positive total associations with the desire for digital avatars of others that were mediated by various motives. For extraverted narcissism, the motivation to be there for others mediated its association with the desire for digital avatars of others. In contrast, antagonistic narcissism and communal narcissism had indirect associations with the desire for digital avatars of others through fear of death for self via the motivation for eternal life, and the motivation to be there for others. Unlike the other narcissistic personality traits, neurotic narcissism did not have a significant total association with the desire for digital avatars of others. However, it did have contrasting indirect associations with this desire. That is, neurotic narcissism had a negative indirect association with the desire for digital avatars of others through the motivation for eternal life, whereas it had a positive indirect association with this desire through the fear of death for others. Neurotic narcissism also had positive indirect associations with the desire for digital avatars of others through the fear of death for self via the motivation for eternal life, and the motivation to be there for others.

It is important to note that we also conducted preliminary analyses in which we included whether participants reported losing a family member or friend during recent years as well as the extent to which they were engaged in online activities. Although these experiences were related to some of our primary variables (e.g., individuals who were more engaged in online activities reported a stronger desire to have a digital avatar of themselves, which aligns with the results of previous research [31]), the inclusion of these experiences in the analyses did not significantly alter the reported results. Consequently, we did not include these variables in the final analyses in the interest of parsimony.

## 4. Discussion

The results were somewhat consistent with our predictions. Extraverted narcissism, antagonistic narcissism, and communal narcissism were positively associated with fear of death (for self and others). Moreover, the motivation for eternal life and the motivation to be there for others had the strongest links with the desire for digital avatars of self or others, and the fear of death for others was positively associated with the desire for digital avatars of others. These findings are consistent with the notion that narcissistic individuals typically exhibit a grandiose sense of self-importance, a desire for admiration, and a lack of empathy for others, and that these characteristics may serve as defense mechanisms to protect against underlying feelings of vulnerability and existential fear, including the fear of death. This pattern of results suggests that the levels of fear of death and the motives for eternal life and being there for others may help to explain the relatively high levels of desire for symbolic immortality reported by those with narcissistic personality traits.

The results for the motivation for a legacy were somewhat surprising because it did not have a unique association with the desire for digital avatars of self or others after controlling for the other predictors in the model. This finding might reflect the idea that legacy is personal, flexible, and more universal than generativity and that it removes the stigmatization of narcissism that applies to the drive to be remembered and to leave a mark. However, selflessness is not the primary drive of legacy, as it is postulated to be in generativity. Narcissism and generativity have an ambiguous relationship with one another [67], and individuals are not acting in a truly generative manner if they are interested in any sort of egocentric result such as leaving a mark or being remembered (i.e., generativity is implicitly linked with altruism; [68]). In contrast, legacy does not include a negative connotation of wanting to be remembered or thinking of oneself [69].

Although neurotic narcissism was similar to the other narcissistic personality traits in terms of its associations with the fear of death for self and the fear of death for others, it diverged from the other aspects of narcissism with regard to its associations with the motives for having digital avatars and the desire to have digital avatars of self or others. That is, extraverted narcissism, antagonistic narcissism, and communal narcissism were positively correlated with each of these motives and desires, whereas neurotic narcissism was not correlated with these motives or desires. This finding might be explained by the fact that *neuroticism*—a basic personality trait that serves as the foundation for neurotic narcissism [7]—has consistent connections with the fear of death [70,71,72,73]. Individuals with elevated levels of neurotic narcissism may have other strategies for attempting to manage their death-related anxiety beyond the sorts of approaches examined in the present study. For example, previous studies have shown that highly neurotic individuals tend to avoid physical sensations—including pleasurable sensations—following reminders of their own eventual deaths [74]. This suggests that individuals with high levels of neurotic narcissism may be so vulnerable to the negative affective experiences surrounding thoughts of their own death that they are forced to find novel strategies to escape these unpleasant emotional states (e.g., avoidance of death-related thoughts).

Although extraverted narcissism, antagonistic narcissism, and communal narcissism were positively associated with the desire for digital avatars of self and others, the mechanisms underlying these associations appeared to differ. This is important because it aligns with the results of previous studies showing that these narcissistic personality traits often diverge in their associations with various outcomes. Results of the parallel mediational models indicated that extraverted narcissism was associated with the desire for digital avatars for self and others through the motivation to be there for others. In contrast, antagonistic narcissism was associated with the desire for these digital avatars through the fear of death for self and the motivation for eternal life and also had an indirect association with the desire for digital avatars of others through the fear of death for self and the motivation to be there for others. Communal narcissism had similarities to both extraverted narcissism (indirect associations with these desires through the motivation to be there for others) and antagonistic narcissism (indirect associations with these desires through the fear of death via the motives for eternal life and being there for others). The results regarding the indirect associations that narcissistic personality traits had with the desire for symbolic immortality through the motivation for eternal life are consistent with the claim that narcissistic characteristics may serve as defense mechanisms to protect against underlying feelings of vulnerability and existential fear, including the fear of death. The findings regarding the indirect associations that narcissism had with the desire for symbolic immortality through the motivation for being there for others are consistent with studies showing that narcissists may engage in various behaviors to maintain a sense of control and avoid confronting their mortality. Thus, being there for others via digital avatars may include a desire to symbolically—and narcissistically—exert some degree of “control” over the future.

The fact that the pattern of results for communal narcissism was similar to the associations that emerged for extraverted narcissism and antagonistic narcissism is in line with the argument that agentic and communal narcissists share the same self-motives for grandiosity, esteem, entitlement, and power and control, but they deploy different means to pursue these self-motives [57]. That is, agentic narcissists deploy agentic means for achieving their goals, whereas communal narcissists tend to rely more heavily on communal means for accomplishing their goals. This suggests that narcissistic individuals are often trying to satisfy similar motives, even if communal narcissists attempt to do so by ostensibly focusing on the needs of others.

The fear of death for self was positively associated with the desire for a digital avatar of self, but it was surprising that this association was not stronger. One possible explanation is that issues concerning the denial of death could have played a role in the relatively modest associations that the fear of death had with the other variables in this study. This possibility is consistent with the idea that a wide range of defensive processes—including denial—protect us from the potentially debilitating anxiety that is involved in the true consideration of our own mortality and may impact the personal meaning we assign to our lives [46,75,76]. It may be beneficial for future studies concerning this topic to attempt to account for the denial of death. For example, some studies have shown that the effects of the fear of death are often moderated by the extent to which individuals try to avoid thinking about their own deaths [77]. The denial of death may also moderate the associations that narcissistic personality traits and the fear of death have with the desire for digital avatars of self and others.

The present results revealed that there were both similarities and differences in the connections that narcissistic personality traits had with the desire for symbolic immortality for self and others. For example, antagonistic narcissism, neurotic narcissism, and communal narcissism had similar indirect associations with the desire for a digital avatar of self through the fear of death for self and the motivation for eternal life, whereas extraverted narcissism was primarily connected with the desire for a digital avatar of self through the desire to be there for others. These results suggest that the fear of death and specific motives to have a digital avatar may play essential roles in understanding the divergent associations that narcissistic personality traits have with the desire to have these sorts of avatars. Further, this suggests that the fear of death and motives concerning control (e.g., eternal life) may play a crucial role in determining the extent to which people—and individuals with narcissistic personality traits in particular—feel about access to this sort of “digital afterlife.” Further, platforms that foster an opportunity for individuals to perceive that they are controlling the future for themselves and those close to them may allow another avenue for denying the reality of death. This may be particularly important for individuals with narcissistic tendencies who may be particularly sensitive to issues surrounding control. It is important to note that the motives behind the desire for digital avatars for self and others may be complex, and it is unlikely that the present study failed to capture some of the motives that may be important. It would be beneficial for future research on this topic to consider an even wider array of motives that may play a role in the desire for digital avatars.

The juxtaposition of the motives for an avatar of self and avatars for others reveals minor differences between the two. The associations between variables are very similar, to a large extent; however, the desire for digital avatars for others demonstrates some additional indirect associations. In other words, there are alternative explanations for the associations when it comes to the desire for digital avatars for others, namely for neurotic narcissism.

One limitation of this research was that we used cross-sectional data to examine mediational hypotheses. Although the results supported narcissistic personality traits having indirect associations with the desire for digital avatars through the fear of death and specific motives, our reliance on cross-sectional data prevents us from concluding the underlying causal processes. For example, a chronic fear of death may lead to the development of certain narcissistic personality traits, rather than narcissism influencing the extent to which an individual fears death. As a result, future research should try to gain a better understanding of the causal links between these variables.

Another limitation was that we relied on participants from a single culture (i.e., Israel). As a result, we do not know whether the present results would generalize to other cultures. However, the present results are similar in some ways to those of previous studies that have examined the associations between narcissism and the fear of death in other cultures [48,49], so it seems unlikely that these results would be limited to Israel. However, most of our participants were Israeli Jews, who may have their own specific attitudes toward death and engagement with the dead (e.g., death may be ascribed to a sacred status). Although there is a pervasive belief in an afterlife across cultures and religions [31], it is unclear how many Israeli Jews actually subscribe to Jewish beliefs concerning the afterlife. Thus, the results observed in the present study could differ across cultures and religions with different attitudes toward death and ways of achieving a semblance of immortality. As a result, it would be extremely helpful for additional studies to examine these issues across various cultures and religions.

An interesting direction for future research in this area would be to consider the nature of the digital avatar that individuals with narcissistic personality traits would choose to employ. For example, research has examined the extent to which users create avatars that accurately reflect their owners or choose to accentuate certain desirable traits [78]. It would be interesting to examine whether narcissistic individuals would develop avatars that accurately represented themselves (e.g., physical characteristics, personality traits)—or at least accurately represented their perceptions of themselves—or knowingly accentuated the desirable qualities of the digital avatar.

## 5. Conclusions

Despite the limitations of this research, we believe it has expanded the current understanding of the connections that narcissistic personality traits have with the desire for symbolic digital immortality through perceived fear of death and specific motives. More specifically, narcissistic personality traits were associated with fear of death and with the desire for symbolic immortality (having an avatar for self and others), with mainly indirect associations via fear of death and the motives for eternal life and to be there for others. These results provide additional support for distinguishing between different aspects of narcissism when considering the connections that narcissism has with the fear of death, motives for immortality, and the desire for particular types of symbolic immortality.

## Figures and Tables

**Figure 1 ijerph-20-06632-f001:**
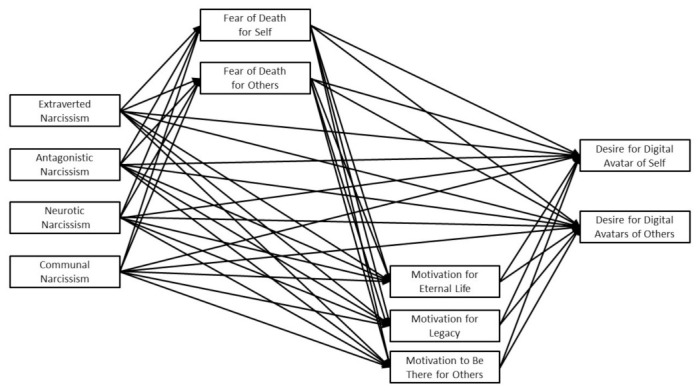
The Proposed Model.

**Table 1 ijerph-20-06632-t001:** Zero-order correlations and descriptive statistics.

	1	2	3	4	5	6	7	8	9	10	11
1. Extraverted Narcissism	—										
2. Antagonistic Narcissism	0.38 ***	—									
3. Neurotic Narcissism	−0.15 ***	−0.04	—								
4. Communal Narcissism	0.48 ***	0.28 ***	−0.18 ***	—							
5. Fear of Death for Self	0.06 *	0.13 ***	0.19 ***	0.15 ***	—						
6. Fear of Death for Others	0.09 **	0.17 ***	0.25 ***	0.13 ***	0.33 ***	—					
7. Motivation for Eternal Life	0.07 *	0.21 ***	−0.03	0.24 ***	0.33 ***	0.13 ***	—				
8. Motivation for Legacy	0.25 ***	0.14 ***	−0.01	0.29 ***	0.29 ***	0.11 ***	0.41 ***	—			
9. Motivation to Be There for Others	0.17 ***	0.10 **	0.02	0.23 ***	0.24 ***	0.10 ***	0.44 ***	0.63 ***	—		
10. Desire for Avatar of Self	0.10 **	0.15 ***	−0.05	0.22 ***	0.23 ***	0.12 ***	0.65 ***	0.43 ***	0.55 ***	—	
11. Desire for Avatars of Others	0.08 *	0.13 ***	0.01	0.20 ***	0.27 ***	0.17 ***	0.58 ***	0.40 ***	0.55 ***	0.79 ***	—
Mean	3.38	2.42	3.12	4.00	2.88	4.04	2.54	4.33	4.28	2.94	3.13
Standard Deviation	0.57	0.49	0.70	1.05	1.05	0.60	2.04	2.17	2.22	2.19	2.30

* *p* < 0.05; ** *p* < 0.01; *** *p* < 0.001.

**Table 2 ijerph-20-06632-t002:** Results for the desire for a digital avatar for self.

	Extraverted Narcissism	Antagonistic Narcissism	Neurotic Narcissism	Communal Narcissism
*Associations with Primary Mediators*				
Narcissism → Fear of Death for Self (FDS)	0.06 *	0.13 ***	0.19 ***	0.15 ***
Narcissism → Fear of Death for Others (FDO)	0.09 **	0.17 ***	0.25 ***	0.12 ***
*Associations with Secondary Mediators*				
Narcissism → Motivation for Eternal Life (MEL)	0.05	0.17 ***	−0.11 ***	0.19 ***
FDS → MEL	0.32 ***	0.31 ***	0.34 ***	0.30 ***
FDO → MEL	0.02	0.00	0.04	0.01
Narcissism → Motivation for Legacy (ML)	0.23 ***	0.11 ***	−0.07 *	0.25 ***
FDS → ML	0.28 ***	0.28 ***	0.30 ***	0.26 ***
FDO → ML	0.00	0.00	0.03	0.00
Narcissism → Motivation to Be There for Others (MBTO)	0.15 ***	0.06 *	−0.04	0.19 ***
FDS → MBTO	0.22 ***	0.22 ***	0.23 ***	0.20 ***
FDO → MBTO	0.02	0.02	0.04	0.01
*Associations with Outcomes*				
Narcissism → Desire for Avatar of Self (Total)	0.10 **	0.15 ***	−0.05	0.22 ***
Narcissism → Desire for Avatar of Self (Direct)	0.00	0.01	−0.04	0.02
FDS → Desire for Avatar of Self	−0.02	−0.02	−0.02	−0.02
Narcissism → FDS → Desire for Avatar of Self	0.00	0.00	0.00	0.00
FDO → Desire for Avatar for Self	0.02	0.02	0.03	0.02
Narcissism → FDO → Desire for Avatar of Self	0.00	0.00	0.01	0.00
MEL → Desire for Avatar of Self	0.50 ***	0.50 ***	0.49 ***	0.50 ***
Narcissism → MEL → Desire for Avatar of Self	0.03	0.08 ***	−0.05 ***	0.10 ***
Narcissism → FDS → MEL → Desire for Avatar of Self	0.01	0.02 ***	0.03 ***	0.02 ***
Narcissism → FDO → MEL → Desire for Avatar of Self	0.00	0.00	0.01	0.00
ML → Desire for Avatar of Self	0.04	0.03	0.03	0.03
Narcissism → ML → Desire for Avatar of Self	0.01	0.00	0.00	0.01
Narcissism → FDS → ML → Desire for Avatar of Self	0.00	0.00	0.00	0.00
Narcissism → FDO → ML → Desire for Avatar of Self	0.00	0.00	0.00	0.00
MBTO → Desire for Avatar of Self	0.31 ***	0.31 ***	0.31 ***	0.31 ***
Narcissism → MBTO → Desire for Avatar of Self	0.05 ***	0.02	−0.01	0.06 ***
Narcissism → FDS → MBTO → Desire for Avatar of Self	0.00	0.00	0.01 ***	0.01 ***
Narcissism → FDO → MBTO → Desire for Avatar of Self	0.00	0.00	0.00	0.00

* *p* < 0.05; ** *p* < 0.01; *** *p* < 0.001.

**Table 3 ijerph-20-06632-t003:** Results for the desire for digital avatars of others.

	Extraverted Narcissism	Antagonistic Narcissism	Neurotic Narcissism	Communal Narcissism
*Associations with Primary Mediators*				
Narcissism → Fear of Death for Self (FDS)	0.06 *	0.13 ***	0.19 ***	0.15 ***
Narcissism → Fear of Death for Others (FDO)	0.09 **	0.17 ***	0.25 ***	0.12 ***
*Associations with Secondary Mediators*				
Narcissism → Motivation for Eternal Life (MEL)	0.05	0.17 ***	−0.11 ***	0.19 ***
FDS → MEL	0.32 ***	0.31 ***	0.34 ***	0.30 ***
FDO → MEL	0.02	0.00	0.04	0.01
Narcissism → Motivation for Legacy (ML)	0.23 ***	0.11 ***	−0.07 *	0.25 ***
FDS → ML	0.28 ***	0.28 ***	0.30 ***	0.26 ***
FDO → ML	0.00	0.00	0.03	0.00
Narcissism → Motivation to Be There for Others (MBTO)	0.15 ***	0.06 *	−0.04	0.19 ***
FDS → MBTO	0.22 ***	0.22 ***	0.23 ***	0.20 ***
FDO → MBTO	0.02	0.02	0.04	0.01
*Associations with Outcomes*				
Narcissism → Desire for Avatar of Others (Total)	0.08 **	0.13 ***	0.01	0.20 ***
Narcissism → Desire for Avatar of Others (Direct)	−0.02	0.00	−0.01	0.01
FDS → Desire for Avatar of Others	0.03	0.03	0.03	0.03
Narcissism → FDS → Desire for Avatar of Others	0.00	0.00	0.01	0.00
FDO → Desire for Avatar of Others	0.07 **	0.07 **	0.07 **	0.07 **
Narcissism → FDO → Desire for Avatar of Others	0.01	0.01	0.02 **	0.01
MEL → Desire for Avatar of Others	0.41 ***	0.41 ***	0.41 ***	0.41 ***
Narcissism → MEL → Desire for Avatar of Others	0.02	0.07 ***	−0.04 ***	0.08 ***
Narcissism → FDS → MEL → Desire for Avatar of Others	0.01	0.02 ***	0.03 ***	0.02 ***
Narcissism → FDO → MEL → Desire for Avatar of Others	0.00	0.00	0.00	0.00
ML → Desire for Avatar of Others	−0.02	−0.02	−0.02	−0.02
Narcissism → ML → Desire for Avatar of Others	0.00	0.00	0.00	−0.01
Narcissism → FDS → ML → Desire for Avatar of Others	0.00	0.00	0.00	0.00
Narcissism → FDO → ML → Desire for Avatar of Others	0.00	0.00	0.00	0.00
MBTO → Desire for Avatar of Others	0.36 ***	0.36 ***	0.36 ***	0.36 ***
Narcissism → MBTO → Desire for Avatar of Others	0.06 ***	0.02	−0.01	0.07 ***
Narcissism → FDS → MBTO → Desire for Avatar of Others	0.01	0.01 ***	0.02 ***	0.01 ***
Narcissism → FDO → MBTO → Desire for Avatar of Others	0.00	0.00	0.00	0.00

* *p* < 0.05; ** *p* < 0.01; *** *p* < 0.001.

## Data Availability

The data presented in this study are available on request from the corresponding authors.

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
