# Peer review of "Who Wants to (Digitally) Live Forever? The Connections That Narcissism Has with Motives for Digital Immortality and the Desire for Digital Avatars"

_ijerph, 2023, doi:10.3390/ijerph20176632_

Round 1

Reviewer 1 Report

I have limited additional comments to contribute to the manuscript. The authors have investigated whether different types of narcissism are related to the desire to live on in digital form (i.e., digital avatars). Mediator variables such as fear of death and different motivations between the independent variables (i.e., different types of narcissism) and the dependent variables were examined. The results largely confirm expectations that individuals with high narcissism are more likely to want to continue living in digital form compared to individuals with low narcissism. As the authors write, two weaknesses of the research are (1) the mediation model based on data from cross-sectional research, which is not suitable to reveal causal relationships. Although the model outlined by the authors is logical, these variables may have other types of relationships with each other; for example, fear of death is just as likely to make someone narcissistic as narcissism is to lead to fear of death, and so on (2) the sample was culturally homogeneous, while the authors begin their study by stating that the inevitable nature of death is a universal challenge to people, to which people seek to respond both individually and as a group. The study variables' relationship may differ depending on the group’s/culture's view of death. I agree with the authors that it would be extremely useful for future research to explore the relationship between these variables in other cultural contexts. The study is well-written, produces interesting results, the sample is adequate for the questions asked, and the statistical methods are sound. I do not know how the above-mentioned problematic aspects of the research could be corrected, so I cannot make any requests to the authors regarding these. Perhaps it is worth revisiting the whole text again for point (1) to keep causal language in the manuscript as limited as possible (or even eliminate it). Thank you for the opportunity to read this interesting paper; I wish the authors good luck!

Author Response

Reviewer 1 Comment 1: “I have limited additional comments to contribute to the manuscript. The authors have investigated whether different types of narcissism are related to the desire to live on in digital form (i.e., digital avatars). Mediator variables such as fear of death and different motivations between the independent variables (i.e., different types of narcissism) and the dependent variables were examined. The results largely confirm expectations that individuals with high narcissism are more likely to want to continue living in digital form compared to individuals with low narcissism.”

Response: We would like to thank the reviewer for their kind words and positive evaluation of our manuscript.

Reviewer 1 Comment 2: “As the authors write, two weaknesses of the research are (1) the mediation model based on data from cross-sectional research, which is not suitable to reveal causal relationships. Although the model outlined by the authors is logical, these variables may have other types of relationships with each other; for example, fear of death is just as likely to make someone narcissistic as narcissism is to lead to fear of death, and so on.”

Response: We completely agree about the limitation concerning our use of cross-sectional data for a mediational analysis. We have tried to be very clear about this limitation.

Reviewer 1 Comment 3: “(2) the sample was culturally homogeneous, while the authors begin their study by stating that the inevitable nature of death is a universal challenge to people, to which people seek to respond both individually and as a group. The study variables' relationship may differ depending on the group’s/culture's view of death. I agree with the authors that it would be extremely useful for future research to explore the relationship between these variables in other cultural contexts.”

Response: We sincerely hope that other researchers will examine these associations in other cultures because it would be very beneficial for scholars to gain a better understanding regarding the extent to which these patterns generalize across cultures.

Reviewer 1 Comment 4: “The study is well-written, produces interesting results, the sample is adequate for the questions asked, and the statistical methods are sound. I do not know how the above-mentioned problematic aspects of the research could be corrected, so I cannot make any requests to the authors regarding these. Perhaps it is worth revisiting the whole text again for point (1) to keep causal language in the manuscript as limited as possible (or even eliminate it).

Response: We have reviewed the manuscript and we do not believe there were any places where we had inadvertently used causal language.

Reviewer 1 Comment 5: “Thank you for the opportunity to read this interesting paper; I wish the authors good luck!”

Response: Thank you for your kind comments and helpful suggestions!

Reviewer 2 Report

I appreciate it very much for having the opportunity to review this manuscript. This study examined an interesting issue, and I have some suggestions:

(1)   "1.1 The Development of Digital Avatars" this part should be re-orginized to stress the outcome variable - Desire for Digital Avatars; at the same time, how is digital immortality different from or same with Digital Avatars? this shoudl also be clearly stated;

(2)the reasons why you focused on Narcissism should be clearly discussed? and the four forms of Narcissism seems to lack of necessary basis (especially two measurements were adopted in this study);

(3) the theoretical and emperical evidences supporting the mediating model should be expanded; especially regarding fear of death;

(4) the reasons why you distinguish the self and others also needed further explanation;

(5)the discussion should be expanded, focusing on the differences of  different forms of Narcissism, as well as the differences between desire for avatar of self and others.

And I look forward to reading the revised version.

Author Response

Reviewer 2 Comment 1: “‘1.1 The Development of Digital Avatars’ this part should be re-organized to stress the outcome variable - Desire for Digital Avatars; at the same time, how is digital immortality different from or same with Digital Avatars? this should also be clearly stated;”

Response: We have revised the manuscript in order to make this clearer. We explore various forms of digital commemoration and distinguish between one-way communication sites and two-way communication applications where interaction with avatars takes place. We further distinguish between avatars created by the bereaved and avatars created with cooperation and collaboration of the to-be-dead. We view the use of digital avatars as being one way in which individuals can acquire a form of digital immortality. We have tried to make this clearer in the revised manuscript.

Reviewer 2 Comment 2: “the reasons why you focused on Narcissism should be clearly discussed? and the four forms of Narcissism seems to lack of necessary basis (especially two measurements were adopted in this study)”

Response: We have added some material to the revised manuscript in order to make it clearer why we thought it was important to use a multidimensional perspective for narcissism.

Reviewer 2 Comment 3: “the theoretical and empirical evidences supporting the mediating model should be expanded; especially regarding fear of death;”

Response: We have included additional information concerning why we were interested in the fear of death as a potential mediator.

Reviewer 2 Comment 4: “the reasons why you distinguish the self and others also needed further explanation;”

Response: We added an extended explanation for the distinction between self and other in relation to narcissism and fear of death.

Reviewer 2 Comment 5: “the discussion should be expanded, focusing on the differences of  different forms of Narcissism, as well as the differences between desire for avatar of self and others.”

Response: We have included some additional information to the discussion that addresses some of the similarities and differences between the narcissistic personality traits as well as the desires for an avatar of the self vs. the desire for avatars of others.

Reviewer 2 Comment 6: “And I look forward to reading the revised version.”

Response: We really appreciated your helpful feedback and suggestions!

Round 2

Reviewer 2 Report

I appreciate it very much for your revision and response. And I still have some minor suggestions:

(1) I suggest you to further re-orginize the manuscript - first, the mian theme  Digital Avatar could be directly introducted in the first sentence, rather than stating the Death rerkated information; 1.1 The Development of Digital Avatars should be streamlined and re-orgnized, for example, the  relation Narcissism and the Desire for Digital Immortality should be pointed our firstly and then  the underlying meding roles , thediscussion on the meiating model should be further strengthened;

(2) the differences between  desire for  Digital Avatar of self and others shoudl be clarly described in the introduction and discussion;

(3)I suggest the aothors to adopted the Mplus to test the integrated model wuth both two outcome variables in the one model;

(4) I also suggest the author to discuss the  desire for  Digital Avatar from the perspective of self-expansion, here are some references - Vasalou, A. & Joinson, A. N. Me, myself and I: The role of interactional context on self-presentation through avatars. Comput.

Hum. Behav. 25, 510–520 (2009); Niu, G. F., Yao, L. S., Kong, F. C., Luo, Y. J., Duan, C. Y., Sun, X. J., & Zhou, Z. K. (2020). Behavioural and ERP evidence of the self‑advantage of online self‑relevant information. Scientific Reports, 10, 20515, https://doi.org/10.1038/s41598-020-77538-5

Author Response

Reviewer 2 Comment 1: “I suggest you to further re-organize the manuscript - first, the main theme Digital Avatar could be directly introduced in the first sentence, rather than stating the Death related information.”

Response: We have re-organized the manuscript following your advice. For example, the idea of digital avatars is now introduced in the first sentence of the manuscript.

Reviewer 2 Comment 2: “The Development of Digital Avatars should be streamlined and re-organized, for example, the relation Narcissism and the Desire for Digital Immortality should be pointed our firstly and then the underlying mediating roles.”

Response:  We have re-organized the “Development of Digital Avatars” and “Narcissism and the Desire for Digital Immortality” sections based on your suggestions. For example, we introduce the idea of narcissism earlier in the “Development of Digital Avatars” section in the revised manuscript. We believe that these changes better reflect existing and relevant knowledge about digital immortality and the intersection of between death, immortality and digital technologies. We further contend that these revisions stress the novelty of the study and its focus on personality traits and namely narcissism, which haven’t received theoretical and empirical attention until now.

Reviewer 2 Comment 3: “the discussion on the mediation model should be further strengthened.”

Response: We have provided additional information about the mediation model that we decided to use for this study.

Reviewer 2 Comment 4: “the differences between desire for Digital Avatar of self and others should be clearly described in the introduction and discussion.”

Response: We thank the reviewer for suggesting this addition. We added the requested discussion in both the introduction and the discussion.

Reviewer 2 Comment 5: “I suggest the authors to adopt the Mplus to test the integrated model with both two outcome variables in the one model.”

Response: We conducted a model similar to the one that was recommended. That is, we used residualized versions of both outcome variables that controlled for the other outcome variable (e.g., we created a variable that represented the desire for a digital avatar of the self that had no correlation with the desire for digital avatars of others). The results of these analyses were similar to the original results that were reported in the manuscript. However, we decided to use the original results due to concerns about the problems with using residualized variables which is sometimes referred to as the “perils of partialling” (e.g., Lynam et al., 2006; Sleep et al., 2017).

Reviewer 2 Comment 6: “I also suggest the author to discuss the desire for Digital Avatar from the perspective of self-expansion, here are some references –

Vasalou, A. & Joinson, A. N. Me, myself and I: The role of interactional context on self-presentation through avatars. Comput.

Hum. Behav. 25, 510–520 (2009); Niu, G. F., Yao, L. S., Kong, F. C., Luo, Y. J., Duan, C. Y., Sun, X. J., & Zhou, Z. K. (2020). Behavioural and ERP evidence of the self‑advantage of online self‑relevant information. Scientific Reports, 10, 20515, https://doi.org/10.1038/s41598-020-77538-5.”

Response: Thank you for this excellent suggestion and these helpful references. We have included information concerning self-expansion in the section concerning “Narcissism and the desire for digital immortality.”